# Convolutional-recurrent neural networks approximate diffusion tractography from T1-weighted MRI and associated anatomical context

**Leon Y. Cai**[1]                                 LEON.Y.CAI@VANDERBILT.EDU
[1] *Department of Biomedical Engineering, Vanderbilt University, Nashville, TN, USA*
**Ho Hin Lee**[2]                               HO.HIN.LEE@VANDERBILT.EDU
[2] *Department of Computer Science, Vanderbilt University, Nashville, TN, USA*
**Nancy R. Newlin**[2]                     NANCY.R.NEWLIN@VANDERBILT.EDU
**Cailey I. Kerley**[3]                       CAILEY.I.KERLEY@VANDERBILT.EDU
[3] *Department of Electrical and Computer Engineering, Vanderbilt University, Nashville, TN, USA*
**Praitayini Kanakaraj**[2]         PRAITAYINI.KANAKARAJ@VANDERBILT.EDU
**Qi Yang**[2]                                        QI.YANG@VANDERBILT.EDU
**Graham W. Johnson**[1]             GRAHAM.W.JOHNSON@VANDERBILT.EDU
**Daniel Moyer**[2]                           DANIEL.MOYER@VANDERBILT.EDU
**Kurt G. Schilling**[4,5]                 KURT.G.SCHILLING.1@VUMC.ORG
[4] *Department of Radiology and Radiological Sciences, Vanderbilt University Medical Center, Nashville, TN, USA*
[5] *Vanderbilt University Institute of Imaging Science, Vanderbilt University, Nashville, TN, USA*
**François Rheault**[6]                FRANCOIS.M.RHEAULT@USHERBROOKE.CA
[6] *Department of Computer Science, Université de Sherbrooke, Sherbrooke, Quebec, Canada*
**Bennett A. Landman**[1,2,3,4,5]        BENNETT.LANDMAN@VANDERBILT.EDU

**Editors:** Accepted for publication at MIDL 2023

## Abstract

Diffusion MRI (dMRI) streamline tractography is the gold-standard for *in vivo* estimation of white matter (WM) pathways in the brain. However, the high angular resolution dMRI acquisitions capable of fitting the microstructural models needed for tractography are often time-consuming and not routinely collected clinically, restricting the scope of tractography analyses. To address this limitation, we build on recent advances in deep learning which have demonstrated that streamline propagation can be learned from dMRI directly without traditional model fitting. Specifically, we propose learning the streamline propagator from T1w MRI to facilitate arbitrary tractography analyses when dMRI is unavailable. To do so, we present a novel convolutional-recurrent neural network (CoRNN) trained in a teacher-student framework that leverages T1w MRI, associated anatomical context, and streamline memory from data acquired for the Human Connectome Project. We characterize our approach under two common tractography paradigms, WM bundle analysis and structural connectomics, and find approximately a 5-15% difference between measures computed from streamlines generated with our approach and those generated using traditional dMRI tractography. When placed in the literature, these results suggest that the accuracy of WM measures computed from T1w MRI with our method is on the level of scan-rescan dMRI variability and raise an important question: is tractography truly a microstructural phenomenon, or has dMRI merely facilitated its discovery and implementation?

**Keywords:** tractography, convolutional-recurrent neural networks, diffusion MRI, T1-weighted MRI, white matter, bundles, connectomics

## 1. Introduction

Diffusion MRI (dMRI) streamline tractography is the gold-standard for *in vivo* estimation of white matter (WM) pathways in the brain (Jones, 2010). One key similarity across modern implementations is the reliance on high angular resolution diffusion imaging (HARDI) (Descoteaux, 1999). HARDI acquisitions allow advanced models, like the fiber orientation distribution (FOD), to better capture the orientations of underlying WM fiber populations, allowing streamlines to be more accurately propagated through the brain (Tournier et al., 2004). One key challenge to this paradigm is that HARDI acquisitions are often time-consuming and can suffer from noise issues and as such are rarely collected clinically. Thus, tractography has primarily been limited to those with the expertise to robustly acquire and analyze HARDI. This renders a significant portion of neuroimaging data unavailable for tractography analyses, limiting advances in the understanding of WM.

To address this concern, prior studies have sought to skip the tractography step for common tractography-based analyses, like WM bundle analysis and structural connectomics, producing voxel-based WM segmentations or probabilistic atlases that require reworking of existing tractography workflows (Yeh, 2020; Sporns et al., 2005; Yang et al., 2022; Alemán-Gómez et al., 2022). As such, a "plug-and-play" solution to facilitate arbitrary subject-specific streamline-based tractography analyses without dMRI remains elusive. We seek to fill this gap by learning the streamline propagator itself from T1w MRI. To do so, we present a novel convolutional-recurrent neural network (CoRNN) trained in a teacher-student framework leveraging T1w MRI, anatomical context, and streamline memory.

To provide intuition for this work, we first consider traditional tractography using FODs fit from dMRI with constrained spherical deconvolution (CSD) (Algorithm 1) (Tournier et al., 2004, 2007). FODs are often represented with 45 8th order even spherical harmonic coefficients captured on a rigid grid of voxels, whereas streamlines are represented as a series of points in continuous 3D space (Tournier et al., 2007, 2019). The general idea is that the FOD information at the $i$th point of a streamline, $\bar{x}_i$, allows one to compute $d\bar{x}_i$, or the unit vector representing the direction to the next point, $\bar{x}_{i+1}$. One can then propagate the streamline to $\bar{x}_{i+1} = \bar{x}_i + \gamma d\bar{x}_i$, where $\gamma$ is the step size.

As such, the first consideration is that given $\bar{x}_i$ one must estimate the FOD at $\bar{x}_i$ even if $\bar{x}_i$ does not fall on the grid. Typically, this is achieved by sampling the FOD grid at $\bar{x}_i$ using some form of interpolation. We refer to sampling the grid using trilinear interpolation as the "SAMP" operation. After SAMP to estimate the FOD at $\bar{x}_i$, one must then compute $d\bar{x}_i = f(FOD(\bar{x}_i))$. There exist many propagators algorithms for $f$. We select the SDStream deterministic algorithm (Tournier et al., 2012).

---

**Algorithm 1:** Traditional streamline propagation

1. Given a grid of FODs from dMRI, $FOD = CSD(dMRI)$, and streamline point $\bar{x}_i$, perform SAMP to obtain $FOD(\bar{x}_i)$.
2. Compute $d\bar{x}_i = f(FOD(\bar{x}_i))$ with a streamline propagator, $f$, like SDStream.
3. Propagate streamline to $\bar{x}_{i+1} = \bar{x}_i + \gamma d\bar{x}_i$, where $\gamma$ is the step size.

---

Prior works have modeled this iterative process with recurrent neural networks (RNNs) and shown that the streamline propagator can be a learned function from a grid of dMRI

signals directly, effectively skipping the CSD step (Poulin et al., 2019). Interestingly, we observe that both FODs and dMRI signals are simply grids of imaging features that happen to be modeled or measured, respectively. Given this, we posit that the grid of imaging features can be learned instead: in this case, from T1w MRI with a convolutional neural network (CONV). Thus, we propose a modified streamline propagation algorithm (Algorithm 2).

---

**Algorithm 2:** Proposed streamline propagation

---

1. Given a grid of learned imaging features from T1w MRI, $I = CONV(T1w)$, and streamline point $\bar{x}_i$, perform SAMP to obtain $I(\bar{x}_i)$.
2. Compute $d\bar{x}_i, \bar{h}_i = g\left(I(\bar{x}_i), \bar{h}_{i-1}\right)$ with a learned streamline propagator, $g$, approximated by an RNN with hidden state, $\bar{h}$.
3. Propagate streamline to $\bar{x}_{i+1} = \bar{x}_i + \gamma d\bar{x}_i$, where $\gamma$ is the step size.

---

We note CONV and the RNN can be trained end-to-end since they are linked by SAMP, a differentiable operation. Further, unlike dMRI which provides high contrast within WM, T1w MRI largely detects consistent intensities within it. Thus, to obtain useful signals in WM, we posit that the receptive field for CONV must be large enough to capture WM tissue borders. Further, since prior studies have demonstrated that neural networks can learn to propagate streamlines from dMRI signals, the solution space for the problem at hand is known to exist for dMRI (Poulin et al., 2019). Thus, we design a framework to constrain that the solution space for a "student" T1w MRI network be close to that modeled by an analogous "teacher" network from dMRI.

In summary, we attempt to tackle the challenge of performing tractography when traditional dMRI acquisitions are unavailable. As such, we propose learning the streamline propagator from T1w MRI and associated anatomical context with three primary contributions: we present a novel interpretation of CoRNN which (1) is linked end-to-end through innovative use of trilinear sampling, (2) is trained with a custom teacher-student framework to constrain the solution space of T1w MRI to be close to that of dMRI, and (3) propagates streamlines directly on T1w MRI in a first for the field, facilitating streamline-based, WM bundle and connectomics analyses without dMRI.

## 2. Methods

### 2.1. Data and data preparation

We utilize paired dMRI and T1w MRI from 100 adult participants from the Human Connectome Project (HCP) for training (80), validation (10), and testing (10) (Van Essen et al., 2013). The testing participants have two imaging sessions each with dMRI and T1w MRI, allowing our method to be evaluated against scan-rescan dMRI performance. The dMRI were acquired on a custom 3T Siemens Skyra (Erlangen, Germany) with multishell single-shot echo planar imaging at b = 1000, 2000, and 3000 s/mm$^2$ with 90 directions per shell (TE/TR = 89.5/5520ms) and 6 b = 0 s/mm$^2$ images. All dMRI were susceptibility, motion, and eddy current corrected (Andersson et al., 2003; Andersson and Sotiropoulos, 2016). The T1w MRI were acquired with 3D MPRAGE (TE/TR=2.1/2400ms).

To prepare the T1w MRI, we compute a brain mask and perform N4 bias correction (Fischl, 2012; Tustison et al., 2010). We then compute the anatomical context utilized for the network in three steps. First, a tissue-type mask is computed, providing cortical GM, deep GM, WM, and CSF segmentations (Tournier et al., 2019; Jenkinson et al., 2012). Second, 132 brain regions are computed with the SLANT deep learning framework and grouped into 46 larger regions based on the BrainColor protocol (Huo et al., 2019). Third, 72 WM bundle regions defined by the TractSeg algorithm are computed with the WM learning (WML) framework (Yang et al., 2022; Wasserthal et al., 2018). All the contextual information is one-hot encoded. The bias corrected T1w MRI is normalized to the median value within the brain mask. This produces 123 total features for learning. Finally, we compute a rigid transformation between the T1w MRI and the Montreal Neurological Institute (MNI) common space at 2mm isotropic resolution (Grabner et al., 2006).

To process the dMRI, we first compute a rigid transformation between the T1w MRI and average $b = 0 \, s/mm^2$ image, allowing the brain and tissue-type masks to be utilized for dMRI processing. Then, we fit FODs to the dMRI using the Tournier et al. (2013) response function and CSD (Tournier et al., 2007). Subsequently, we compute 1 million streamlines for the training and validation FODs with MRTrix3 SDStream anatomically constrained tractography via the tissue-type and brain masks, seeding at the GM/WM interface, a step size of 1mm, and default tracking parameters otherwise (Tournier et al., 2019, 2012; Smith et al., 2012). The 1 million streamlines are then randomly split into chunks of 1000 streamlines each. For labels, we compute unit vectors representing the steps between all adjacent points on a streamline in spherical coordinates ($\theta$ and $\phi$). This yields labels that are one point shorter than streamlines, so the last points of all streamlines are dropped.

All data are rigidly moved to 2mm MNI space for training and evaluation.

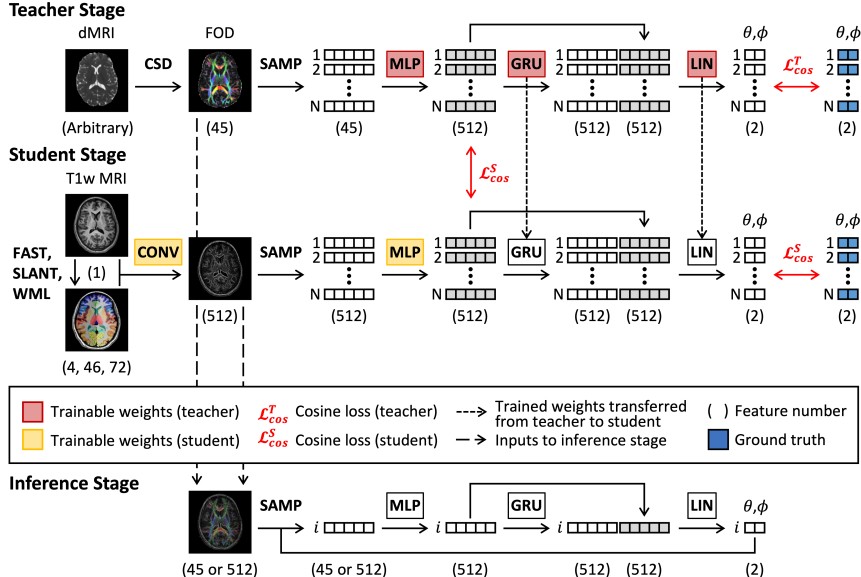

Figure 1: Teacher-student training and inference frameworks.

## 2.2. Teacher-student training and network architectures

We train our approach in a teacher-student framework (Figure 1), learning a streamline propagator from FODs with the teacher before training the student with T1w MRI.

For the teacher, we seek to balance out local information with streamline memory (Poulin et al., 2017). As such, we first use a four-block MLP to embed FOD information queried with SAMP at streamline point locations where each block consists of a linear layer (size 512), a batch normalization layer, and a leaky ReLU activation (slope 0.1) (Ioffe and Szegedy, 2015). The MLP output is then fed into two stacked gated-recurrent-unit (GRU) RNNs (hidden size 512) to encode streamline memory (Cho et al., 2014). The outputs of both are then concatenated and projected with a linear layer (LIN) to the 2D output space representing $d\bar{x}$ as $\theta$ and $\phi$. During training, we use a batch size of 1 on the image level, and a randomly selected batch of 1 chunk or 1000 streamlines for the SAMP operation. We constrain that the network output, $d\bar{x}_{pred}$, be close to the labels, $d\bar{x}_{label}$, with a cosine similarity loss, $\mathcal{L}_{cos}(d\bar{x}_{pred},\ d\bar{x}_{label})$, averaged across all streamlines in the chunk:

$$\mathcal{L}_{cos}(\bar{z}_1, \bar{z}_2) = 1 - \frac{\langle \bar{z}_1, \bar{z}_2 \rangle}{\|\bar{z}_1\|\|\bar{z}_2\|} \tag{1}$$

For the student model, we use a single 7x7x7 3D convolutional projection layer for CONV, providing a 1.4x1.4x1.4cm receptive field in 2mm space. We then utilize SAMP and the same MLP, GRU, and LIN architectures as the teacher to convert the CONV output to streamline points and compute $d\bar{x}_{pred}$. We also utilize frozen weights for the GRU and LIN modules transferred from the teacher. When training the student, we enforce the same constraint against the labels and additionally constrain that the output of the student MLP be similar to that of the teacher MLP with an additional cosine similarity loss.

All models, training, and inference are implemented in PyTorch 1.12.1 and performed on an NVIDIA RTX Quadro 5000 or RTX A6000 (CUDA 11.6). We use an Adam optimizer with a constant learning rate of 0.001 and stop training with no improvement in validation loss after 200 epochs (Kingma and Ba, 2014). The models with the lowest validation loss are used for inference and evaluation (Appendix A).

## 2.3. Seeding, tracking, and stopping during inference

The teacher and student propagate similarly during inference. First, dMRI is converted to FOD with CSD for the teacher and bias corrected T1w MRI and context are embedded with CONV for the student (Tournier et al., 2007). These operations occur only once. Propagation then begins with SAMP to query the image grid at a seed location and initialization of the GRU hidden state to zero. $d\bar{x}$ and the new hidden state are then computed and used to determine the next streamline point location, and so on. We set $\gamma$ to 1mm.

Unlike the training data, we randomly seed points in WM defined by non-zero locations in the tissue-type mask: empirically, this yielded a higher proportion of seeds that became successful streamlines than GM/WM interface seeding. However, seeding in WM requires bidirectional tracking, as unidirectional tracking from the middle of the brain would only capture half of a given WM fiber. Thus, we perform bidirectional tracking by first tracking in one direction until stopping criteria are met, flipping the streamline, performing a forward pass to obtain the new hidden state, and subsequently tracking in the reverse direction.

We implement tracking in batches since, unlike traditional tractography, parallelization of the tracking process with neural networks must occur on the step level as opposed to the streamline level. Thus, all streamlines in a batch are stepped and evaluated for stopping criteria simultaneously. Once all have stopped or been rejected, the reverse tracking process begins. Once reverse tracking is complete, a new batch of streamlines is seeded and the process is repeated until at least 1 million streamlines are generated.

As with the training data, we use anatomically constrained stopping criteria as defined by Smith et al. (2012). Unlike the training data, however, we are unable to use the MRTrix3 implementation due to the neural networks and the modified parallelization. Thus, we reimplement slightly modified but analogous criteria (Appendix B).

## 2.4. Model evaluation

Since each participant in the test set has two dMRI imaging sessions (dMRI 1 and 2) and a T1w MRI, we generate four tractograms each: (1) MRTrix3 tissue-type anatomically constrained SDStream tractography on dMRI 1 with the original stopping criteria to those reimplemented in Appendix B, (2) the same SDStream tractography on dMRI 2, (3) the teacher network on dMRI 1, and (4) the student network on T1w MRI (Tournier et al., 2019; Smith et al., 2012). We call (1) the reference and compare (2)-(4) to it to contextualize our results in the setting of scan-rescan dMRI variability.

We evaluate the different methodologies with two commonly used tractography applications: bundle analysis and structural connectomics (Yeh, 2020; Sporns et al., 2005). For bundles, we identify 39 WM bundles from each tractogram with RecoBundlesX (Appendix C) (Garyfallidis et al., 2018; Rheault, 2020). We compare bundle streamline count, volume, length, span, and surface area between methodologies as well as geometric agreement with the bundle adjacency streamlines distance metric and Dice similarity coefficient (Rheault, 2020; Schilling et al., 2021; Yeh, 2020). For connectomics, we use 97 cortical regions defined by the SLANT BrainCOLOR framework and compute two types of connectomes, one with edges weighted by the streamline count between any two given regions and one weighted analogously by average streamline length (Huo et al., 2019; Sporns et al., 2005). We compare between methodologies with connectome Pearson correlation and the agreement between the maximum modularity, average betweenness centrality, and characteristic path length scalar graph measures as computed with the Brain Connectivity Toolbox (Rubinov and Sporns, 2010). Detailed definitions of all measures are included in Appendix D.

We do not statistically characterize differences between methodologies. Whether differences are significant at N=10 would not drastically change results interpretation as our goal is to understand trends with regards to the dMRI literature to assess framework viability.

## 3. Results

We provide tractograms generated on a representative subject across methodologies in Figure 2a and observe striking similarities overall but noticeably noisier streamlines near the cortex in those generated with SDStream. We posit this is likely due to noisy or small dMRI signals near the GM/WM interface being difficult to reproduce in data-driven paradigms. In either case, we find these tractograms to be appropriate for further investigation.

### 3.1. Bundle evaluation

We plot the median bundle adjacency and Dice coefficients per bundle in Figure 3a (see Appendix E for bundle-specific plots). We find bundle adjacency is higher in both the teacher and student compared to rescan with SDStream, but by less than 0.5mm. We find equivalent Dice across bundles around 0.6-0.7 for all methods. These results suggest good geometric agreement between bundles generated from our models and those from SDStream upon rescan. We find good visual agreement in a representative sample of a bundle with 0.6-0.7 Dice between the student network and the reference in Figure 2b. We plot the median percent difference in streamline count per bundle in Figure 3b and find around 50% fewer streamlines identified per bundle with our models compared to SDStream upon rescan. We plot the median absolute percent difference in characteristics per bundle in Figure 3c and find increasing error from SDStream upon rescan to the teacher to the student across all measures. We also find higher student errors for the voxel-based surface area and volume measures at around 15% and 25%, respectively, but lower errors for the streamline-based length and span measures at around 5-10%. These results suggest that the morphology of student-identified streamlines are in high agreement with SDStream, even if the number or voxel coverage of those streamlines are in lower agreement.

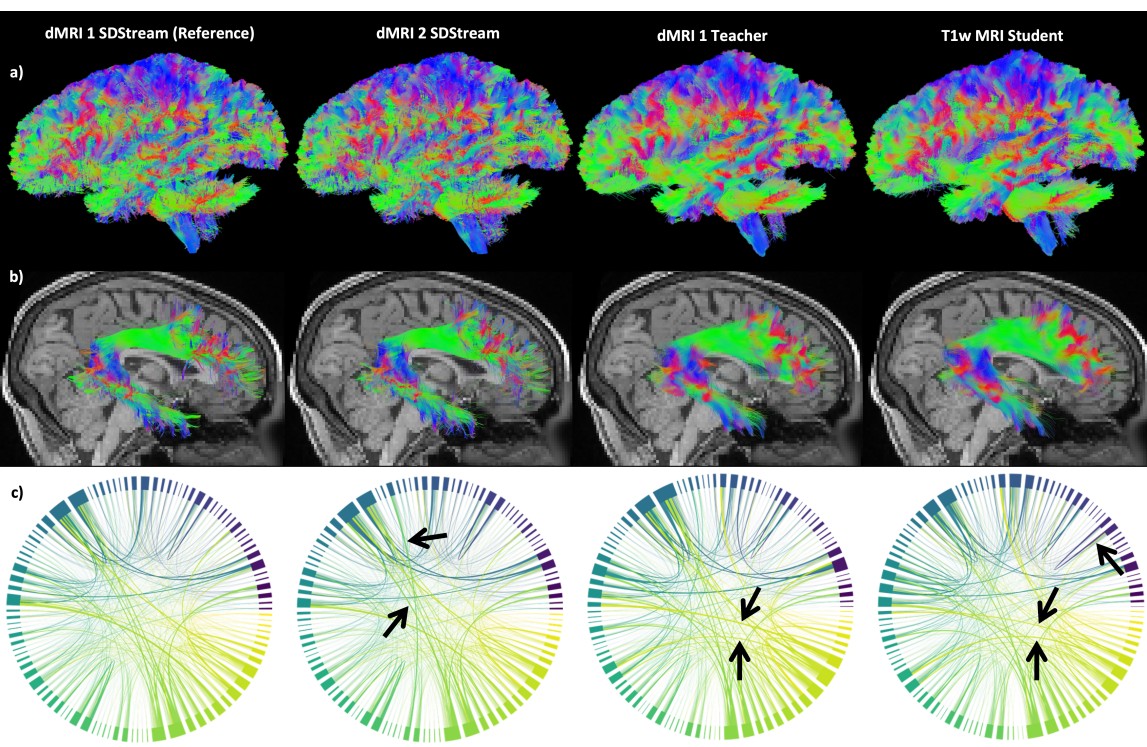

Figure 2: Representative samples by methodology. (a) Left view of tractograms. (b) Right view of the right arcuate fasciculi. (c) Connectomes weighted by streamline count. Arrows denote visually notable differences compared to the reference.

### 3.2. Connectomics evaluation

We plot correlations for the two connectome types in Figure 3d. For connectomes weighted by streamline count, we find decreasing correlations from SDStream upon rescan to the teacher to the student, at around 0.9-1, 0.8-0.9, and 0.7-0.8, respectively. For the connectomes weighted by average streamline length, we find a similar trend with correlations around 0.5, 0.4-0.5, and 0.4-0.5, respectively. As this drop from rescan SDStream to student is more obvious in the connectome weighted by streamline count, we visualize a representative sample in Figure 2c. We observe small differences across methodologies, with few visually obvious differences. Further, we plot differences in maximum modularity, average betweenness centrality, and characteristic path length between methodologies in Figure 3e and find the graph theory measures all have at most 5-10% difference to the reference indicating high agreement, despite reduced connectome correlations with the student network.

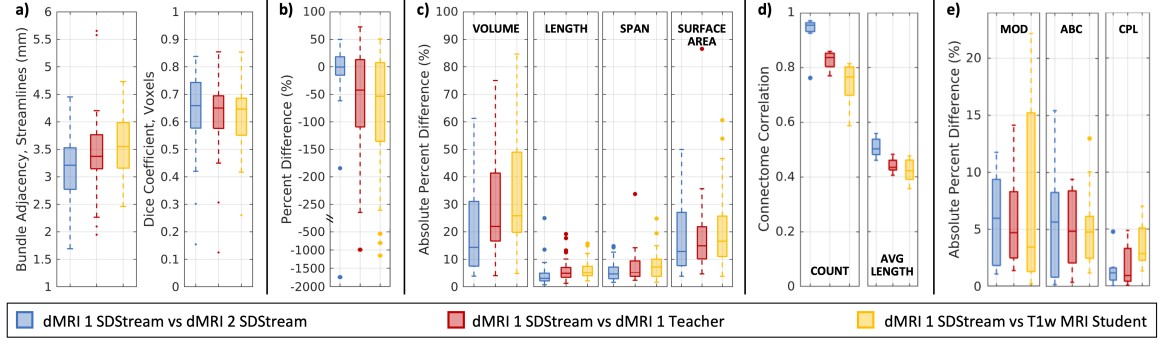

Figure 3: Comparisons of bundle geometry (a), streamline count (b), and characteristics (c) plotted as bundle-wise medians, as well as connectomes (d) and graph measures (e) by methodology.

## 4. Discussion

Prior studies have identified non-negligible scan-rescan variability in dMRI tractography-based measures. One study placed estimates between <5-10% coefficient of variation (CoV) for both bundle and connectomics measures (Cai et al., 2021). Other studies estimated 1-20% CoV for bundles and 5-30% for connectomics (Besseling et al., 2012; Roine et al., 2019). In this work, we quantify the scalar accuracy of tractography-based measures to be around 5-15% absolute difference which translates to approximately 3-10% CoV after accounting for the $\sqrt{2}$ conversion between them. This places the accuracy of our method on the level of variability attributable to scan-rescan effects in traditional dMRI tractography. Notably, this is a contextualization, and not a direct comparison between accuracy and variability.

As this is the first study into CoRNN tractography with T1w MRI, we identify four key limitations. (1) It is unclear whether the network has merely "memorized" propagation directions based on location or if it has truly "learned" to perform tractography on T1w MRI or what it even means to distinguish the two. (2) We did not investigate how other

loss functions or architectures impact performance. (3) We did not pursue a probabilistic framework as neural networks traditionally approximate deterministic functions, though prior studies have shown that probabilistic tractography can be approximated from dMRI with RNNs through reparameterization (Benou and Riklin Raviv, 2019). (4) To assess viability, we only used data from one scanner on 1/10 of the total HCP cohort with one implementation each for bundles and connectomics. Thus, future work should (1) investigate how the model propagates through brains with obvious lesions, (2) more completely explore optimal architectures and designs, (3) study how probabilistic frameworks may be integrated, and (4) further validate the model and framework on external data considering multiple scanners, larger and more diverse cohorts, and different analysis approaches. Future work should also study if the CoRNN framework may allow dMRI harmonization networks to be optimized directly for tractography (Yao et al., 2023; Nath et al., 2019).

Last, the existence of our model raises an important question: is tractography truly a microstructural phenomenon, or is it a reflection of brain shape and macrostructure that has merely been discovered and implemented through dMRI? The answers to this question are potentially broad and paradigm-shifting. Thus, prior to robust investigation and further model validation and optimization, we recognize this framework cannot be an all-purpose replacement for dMRI tractography. Model weights and source code for inference are made available at `github.com/MASILab/cornn_tractography`.

## Acknowledgments

This work was conducted in part using the resources of the Advanced Computing Center for Research and Education at Vanderbilt University, Nashville, TN. This work was supported by the National Institutes of Health (NIH) under award numbers 5R01EB017230, 1U34DK123895-01, U34DK123894-01, P50HD103537, U54HD083211, U54HD083211-S1, K01EB032898, and T32GM007347 and by the National Science Foundation (NSF) under award number 2040462. This research was conducted with the support from the Intramural Research Program of the National Institute on Aging of the NIH. The content is solely the responsibility of the authors and does not necessarily represent the official views of the NIH or NSF.

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

## Appendix A. Loss curves

We include the loss curves for both the teacher and student models in Figure A1. We observe that both the teacher and student converge. Additionally, we observe that the loss curves are noticeably noisy, as expected for a training paradigm where the streamlines are randomly sampled every batch. Last, we also observe that the teacher network training loss increases slightly after convergence, likely due to overfitting to previously randomly sampled streamlines. Regardless, the model weights saved at the point of convergence are used for the experiments.

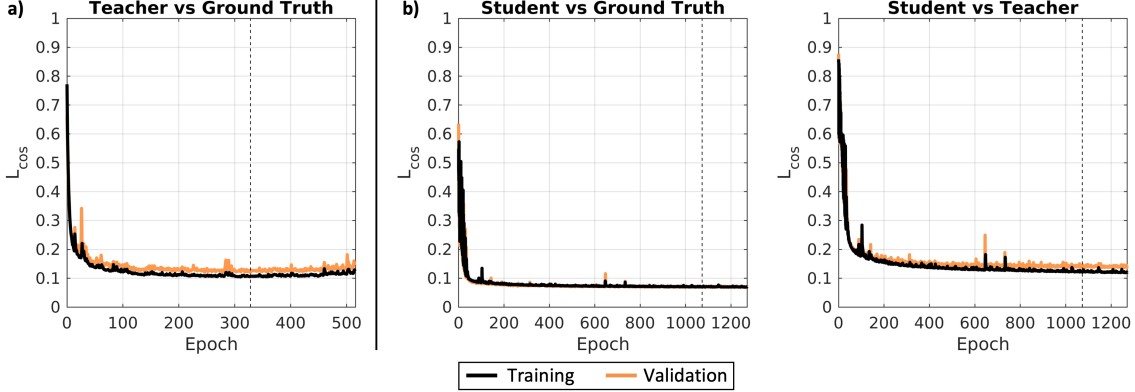

Figure A1: Loss curves for the teacher network (a) and the student network (b). The points of convergence, defined as no improvement in validation loss after 200 epochs, are denoted with dotted lines.

## Appendix B. Reimplementation of anatomically constrained stopping criteria

The following is a reimplementation of the criteria originally described Smith et al. (2012) and implemented in MRTrix3 (Tournier et al., 2019).

For streamlines propagated to the ith point, they are rejected if either of the following criteria are met, where $P_T(\bar{x})$ is the percent contribution of tissue-type $T$ at streamline point location $\bar{x}$, trilinearly interpolated.

1. The streamline enters CSF; defined as $P_{CSF}(\bar{x}_i) > 0.5$ and $P_{CSF}(\bar{x}_i) > P_{CSF}(\bar{x}_{i-1})$; or

2. the streamline takes too sharp of a turn in WM; defined as $P_{WM}(\bar{x}_i) > 0.5$, $P_{WM}(\bar{x}_{i-1}) > 0.5$, and the angle between the current predicted step and the previous predicted step is greater than $60°$.

Streamlines are terminated at the $i$th point if they are not rejected and have either exceeded the maximum length of 250mm or meet any of the following criteria, where CGM is cortical GM, DGM is deep GM, and MASK is a brain mask.

3. The streamline enters CGM; defined as $P_{CGM}(\bar{x}_i) > 0.5$ and $P_{CGM}(\bar{x}_i) > P_{CGM}(\bar{x}_{i-1})$; or

4. the streamline exits the brain mask; defined as $P_{MASK}(\bar{x}_i) > 0.5$; or

5. the streamline takes too sharp of a turn in DGM; defined as $P_{DGM}(\bar{x}_i) > 0.5$, $P_{DGM}(\bar{x}_{i-1}) > 0.5$, and the angle between the current predicted step and the previous predicted step is greater than $60°$; or

6. the streamline exits DGM; defined as $P_{DGM}(\bar{x}_i) < 0.5$ and $P_{DGM}(\bar{x}_{i-1}) > 0.5$.

Unlike the traditional criteria, these implementations do not allow for back-tracking or consideration of streamline morphology beyond the previous streamline point. Additionally, due to the initialization of the hidden state as zero, we provide a buffer of the first five steps where criteria number 2 is ignored to allow streamlines to achieve a more useful hidden state. These five steps are then removed prior to flipping the streamlines for reverse tracking. Last, after reverse tracking, streamlines that do not meet the minimum length requirement of 50mm are discarded.

We note criteria number 6 required empirical tuning to avoid erroneous termination.

## Appendix C.  Bundle definitions

Using the RecoBundlesX tool, we dissect the following bundles from tractograms (Table A1) (Garyfallidis et al., 2018; Rheault, 2020).

Table A1:  Names and abbreviations of bundles investigated with RecoBundlesX. Right and left are investigated separately.

| Abbreviation | Name |
| --- | --- |
| AC | Anterior commissure |
| AF | Arcuate fasciculus |
| CC Fr 1 | Corpus callosum, Frontal lobe (most anterior part) |
| CC Fr 2 | Corpus callosum, Frontal lobe (most posterior part) |
| CC Oc | Corpus callosum, Occipital lobe |
| CC Pa | Corpus callosum, Parietal lobe |
| CC Pr Po | Corpus callosum, Pre/Post central gyri |
| CC Te | Corpus callosum, Temporal lobe |
| CG | Cingulum |
| FAT | Frontal aslant tract |
| FPT | Fronto-pontine tract |
| FX | Fornix |
| ICP | Inferior cerebellar peduncle |
| IFOF | Inferior fronto-occipital fasciculus |
| ILF | Inferior longitudinal fasciculus |
| MCP | Middle cerebellar peduncle |
| MdLF | Middle longitudinal fascicle |
| OR ML | Optic radiation and Meyer's loop |
| PC | Posterior commissure |
| POPT | Parieto-occipito pontine tract |
| PYT | Pyramidal tract |
| SCP | Superior cerebellar peduncle |
| SLF | Superior longitudinal fasciculus |
| UF | Uncinate fasciculus |

## Appendix D. Definitions of WM bundle and connectomics measures

### D.1. Bundle measures

We note that while voxel-based measures allow for comparison of bundle coverage, streamline-based measures provide additional insight into shape that the voxel-based ones do not (Rheault, 2020).

- Bundle adjacency: a streamline-based distance metric representing the minimum pairwise direct-flip distance on bundle centroids in mm (Garyfallidis et al., 2012, 2018).

- Dice coefficient: a voxel-based similarity metric representing the ratio of intersecting over total volume ranging from 0 to 1 (Schilling et al., 2021).

- Streamline count: the number of streamlines identified in the bundle.

- Bundle volume: a voxel-based metric, the volume of the bundle (Yeh, 2020).

- Bundle length: a streamline-based metric, the average length of the streamlines in the bundle (Yeh, 2020).

- Bundle span: a streamline-based metric, the average distance between the ends of streamlines in the bundle (Yeh, 2020).

- Bundle surface area: a voxel-based metric, the surface area covered by the streamlines in the bundle (Yeh, 2020).

### D.2. Connectomics measures

- Connectome Pearson correlation: the Pearson correlation between the edges of two connectomes

- Maximum modularity: the degree to which the nodes of the connectome may be subdivided into separate non-overlapping groups, computed from connectomes with edges weighted by streamline count (Rubinov and Sporns, 2010).

- Average betweenness centrality: the average fraction of shortest paths through the connectome in which the brain regions participate, computed from connectomes weighted by average streamline length (Rubinov and Sporns, 2010).

- Characteristic path length: the average shortest path between brain regions, computed from connectomes weighted by average streamline length (Rubinov and Sporns, 2010).

## Appendix E. Bundle-wise geometric comparisons

We provide Figure A2 to illustrate the geometric agreements by bundle between different methodologies.

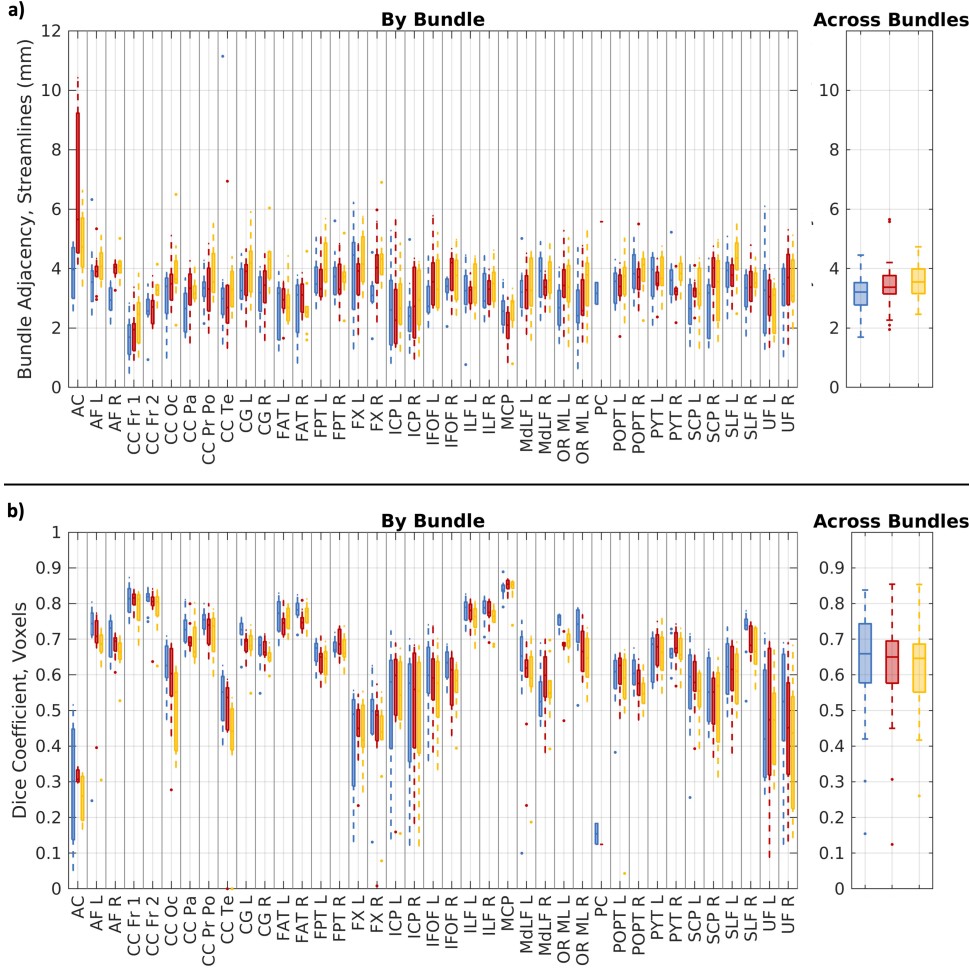

Figure A2: Geometric comparisons of bundles, plotted for each bundle in the left panels and as the distribution of bundle-wise medians in the right. (a) Bundle adjacency, a streamline-based metric, is lower with increasing agreement. (b) The Dice coefficient, a voxel-based metric, is higher with increasing agreement. The right panels are a reproduction of Figure 3a.

## Appendix F. Comparison against non-linear registration of an atlas

To further place our method in the literature, we provide an additional analysis against a common T1w MRI-only method of bundle and connectomics analysis: non-linear registration of streamlines from an atlas. We provide Figure A3 to illustrate the geometric agreement across bundles and the correlations between connectomes generated from non-linear registration of an atlas compared to the methodologies investigated presently. We use the FOD and T1w MRI atlases provided by Lv et al. (2023), computing 1 million streamlines in atlas space with anatomically constrained SDStream tractography as described in section 2.4. The T1w MRI atlas is subsequently non-linearly registered to each subject's T1w MRI and the resultant transform is used to warp the streamlines from atlas space to subject space. Bundles and connectomes are then computed and compared to the reference method following section 2.4. We find improved similarity between our method and the reference compared to registration of the atlas and the reference in all cases, suggesting improved subject-specificity with our method over non-linear registration of atlases.

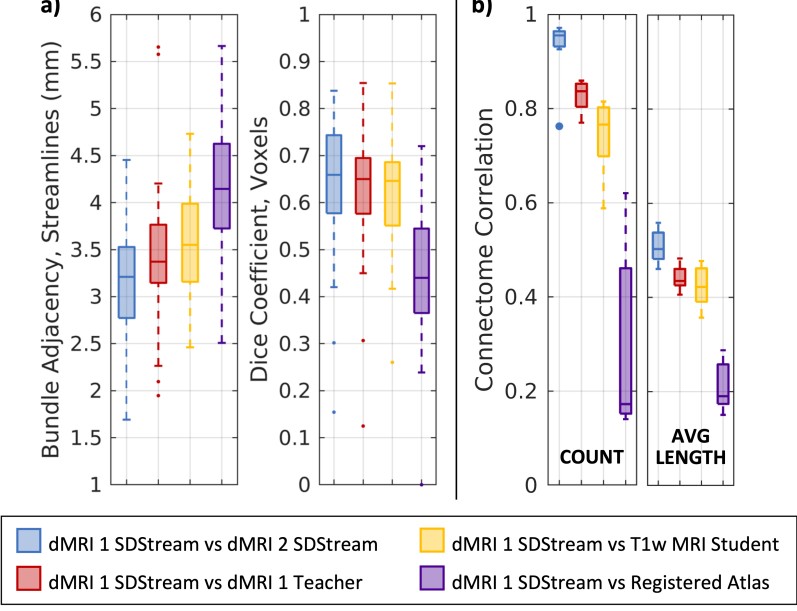

Figure A3: Comparisons of bundle geometry (a) and connectomes (b) by methodology including non-linear registration of an atlas.

