# OpenReview forum: "Convolutional-recurrent neural networks approximate diffusion tractography from T1-weighted MRI and associated anatomical context"
_MIDL.io/2023/Conference — MIDL 2023 Poster_

### Official Review · Reviewer_1Vh6 · 2023-02-04

**Confidence:** 3
**Preliminary Rating:** 3

**Summary:**

This paper proposes a deep learning framework that can perform dWI-like tractography from T1w MRI scans when the corresponding dMRI scan data is unavailable. The model consists of a convolutional-recurrent neural network linked together by trilinear sampling and is end-to-end learnable. The model is trained with a teacher-student learning paradigm to leverage the associated anatomical context from T1w MR and streamline memory from DWI data.

Validation is performed on the Human Connectome Project data and the method is compared against two tractography paradigms, namely WM bundle analysis and structural connectomics. Preliminary results suggests that the method can be applied to T1w MRI for facilitating subject-specific, streamline-based, WM bundle and connectomics analyses without dMRI.

**Strengths:**

1. The methodological novelty along with the potential for application of this framework are major strengths of the contribution and an exciting development for the field. Given that HARDI acquisitions can be time consuming, having a reliable framework that can generate tractography results from a more widely available T1w scan could potentially be very beneficial for multimodal connectomics studies, which tend to have modest dataset sizes. The end-to-end deep learning solution makes it a very attractive alternative for adoption in future applications.

2. The presentation of the paper is excellent. The main motivation and explanation of the methodology and contributions of the work in the context of relevant literature come across very well. The experimental setup is reasonable as a preliminary proof of concept to evaluate the main claims of the work.

**Weaknesses:**

1. The authors do not explicitly mention the computational cost of training/inference with this framework and how it compares against the baselines reported.

2. Validation is performed on a relatively small cohort (10 subjects) with a single data split which makes it difficult to understand whether the performance differences in Fig. 3 against comparative methods are consistent across larger cohorts. Moreover, connectomics comparisons have been performed on a single cortical parcellation (SLANT BrainCOLOR framework), making it hard to comment on the broader applicability of the framework.

**Deanonymize Review:**

no

**Detailed Comments:**

A small suggestion would be to increase the font size, possibly separating the teacher and student networks better to improve the readability of Fig. 1.

**Paper Type:**

methodological development

**Questions To Address In The Rebuttal:**

In my opinion, the major weakness is the scope of the evaluation, especially given the size of the testing cohort and needs further verification on a more reasonably sized cohort from the same scanner. The authors briefly allude to this point within the conclusion/discussion section as a potential limitation, but demonstrating generalizability of the framework (as opposed to a mere memorization of patterns) is an important requirement for a methodological deep learning contribution.

---

### Official Review · Reviewer_EZGy · 2023-02-05

**Confidence:** 3
**Preliminary Rating:** 5
**Recommendation:** Oral

**Summary:**

The submission introduces a novel convolution-recurrent neural network that can facilitate the tractography analysis when dMRI data is not available. CoRNN is built on a teacher-student framework. Hundred subjects from the HCP cohort are used in the experiments. A difference of about 5-15% is found between the outcomes of the here proposed tool and that of traditional dMRI-based tractography methods. The evaluations point to a very promising set of first results.

**Strengths:**

Novelty: novel deep learning tool that allows us to compute WM measures without dMRI close to what would be computed from a rescan; first time subject-specific streamline-based WM bundle and connectomics analysis

The paper is well written and key points are clearly discussed. Good discussion of previous work

The model is evaluated on four methods: dMRI1 tractography; dMRI2 tractography; teacher network on dMRI and student network on T1

Nice Fig 2 making comparison among the various methods qualitatively easy.

If fully validated the impact of the proposed framework could be significant.



**Weaknesses:**

A more complete analysis of the network and its components would be warranted for a full performance review.

A larger test data set would have allowed for a possibly statistically significant analysis.



**Deanonymize Review:**

no

**Paper Type:**

both

**Questions To Address In The Rebuttal:**

Hundred subjects' images are used from HCP for the experiments. Why not more if this subset only allows for 10 subjects to be in the test data set, which is not sufficient for locating statistically significant findings?

Why don't the authors have access to the GM/WM interface seeding during inference? Can you elaborate? The T1-based segmentation of the input should be available.

How starlight forward would it be to implement probabilistic tractography solutions in the proposed framework? As mentioned by the authors as well it has been shown that it is possible to also approximate it via RNNs

---

### Official Review · Reviewer_iTZC · 2023-02-05

**Confidence:** 4
**Preliminary Rating:** 3
**Recommendation:** Poster

**Summary:**

The paper presents a teacher-student deep learning system to extract tractography streamlines from T1w images. The method thus has two components: a teacher network, trained on DWI inputs and a student model trained on T1w inputs. The models are trained on the HCP dataset and predicted held-out subject tractograms are compared with the SDStream (MRtrix) both qualitatively and quantitatively.

**Strengths:**

- The method produces good-looking and promising results.
- With dMRI acquisitions being substantially more scan-time intensive than T1w images, tractogram from T1w images promise "cheaper" tractogram generation.
- The method seems sound and well-designed and is a clear strong point for the approach

**Weaknesses:**

Unclear if teacher-student model is necessary.

it would have been nice to see, if just a model trained on the T1w images already achieves good results. No such results are shown and thus it is unclear, whether the teach-student-design is necessary at all.

Baseline results

Evaluations only include results obtained from DWI input. However, it seems the real challenge is the modality transfer. It would have been nice to see that being included as well, for example using methods such as [1] to convert from T1w to DWI and the running SDStream on top of that. This way, the work is the worst-performer in the list but also does not have a similar competing paradigm to compare against.

Ground truth

The existence of ground truth is a well-known problem for dMRI tracking, but it reappears in this paper as well. The authors have tried to address this issue by using ground truth from dMRI. Although not a critical flaw here, I would recommend establishing some higher-quality ground truth (manual ground truth, see [2] for example). Alternatively, simulated and pohysical phantoms have been used in places to establish higher quality gfround truths.

Limitations of T1w from DWI

As authors point out, it is unclear how well their method would translate to unseen (e.g. lesions) data types. However, extending this limitation it also remains unclear which information is available or not available in synthetic tractograms. It is entirely unclear to me, whether the quality of the obtained tractogram is in fact better than any random (non-linearly co-registered) tractogram of a different subject. Therefore, the value of this synthetic tractogram remains a question in my eyes. Do anatomical/pathological differences make an impact. The question whether tractogram from T1w is fundamentally useful at all remains unanswered in my opinion.

[1] https://link.springer.com/chapter/10.1007/978-3-030-87234-2_50
[2] https://doi.org/10.1007/s00429-020-02129-z

**Deanonymize Review:**

no

**Paper Type:**

methodological development

**Questions To Address In The Rebuttal:**

Most of the weaknesses are beyond the typical scope of the rebuttal process. Personally, I feel the fact that no other tractogram from T1w paradigm was formulated is the core weakness. The question of fundamental limitations of tractography from T1w (i.e. purely anatomically based tractograms) the second weakness. For example, is the result just purely a non-linear registration of another subject, how do anatomical differences impact reultrs etc.

---

### Official Review · Reviewer_gyyV · 2023-02-07

**Confidence:** 4
**Preliminary Rating:** 2

**Summary:**

This work proposes a deep learning-centered method for learning the diffusion tractography propagation function to be applied over a T1w MRI. As stated in this paper, one of the primary limitations of brain tractography-based analyses is the reliance upon diffusion MR images. The proposed methods seek to relax or remove this limitation by learning to re-create the "tracking" of fibers found by classical tractography methods, but using only T1w images as input. A CNN-RNN architecture is proposed, along with a unique student-teacher training regime. The learned tractography propagation method is tested in 10 subjects, and extensive performance metrics are calculated on the predicted fiber tracts, including voxel, streamline, and connectome-based similarity measurements.

**Strengths:**

1. Comparing fiber tractography reconstructions is a long-standing problem in the field, and the authors wisely evaluate the proposed method with a variety of comparison metrics across different levels of abstraction (voxel, streamline, and graph/connectome-based). This allows for a level of clarity and insight that is necessary to interpret the historically unclear results in tractography. In the context of machine learning, it is important to note that these metrics were not explicitly optimized over, which may indicate a level of robustness for the method.

2. The handling of multiple modalities in the proposed method is itself unique and interesting. The student-teacher paradigm, while used in other multi-modal MRI and transfer learning methods, is rarely used in the diffusion literature, and the proposed method applies it organically. The segmentation and labeling of regions in T1w images is also interesting as an input to the student network.

3. The implementation and incorporation of previous methods in this work is laudable, and the release of the source code will be valuable to the tractography community.

**Weaknesses:**

4. Unfortunately, the basic premise of the paper seems questionable - the information necessary for tractography is simply not present in T1w brain MRIs. Indeed, any method that attempts tractography with only T1w seems vulnerable to regression to the "average" brain in the dataset. The motivation for this work, that tractography is limited by its dependence on diffusion MRIs, is a real challenge, but it exists because no other commonplace MR modalities (T1w/MPRAGE as in HCP, T2w, etc.) capture the same directional information as diffusion MRIs. As given in the paper:

> "Further, unlike dMRI which provides high contrast within WM, T1w MRI largely detects consistent intensities within it. Thus, to obtain useful signals in WM, we posit that the receptive field for CONV must be large enough to capture WM tissue borders."

This first sentence agrees with what is stated above, and admits a strong distinction between T1w image features and dMRI features. The second sentence and the rest of the proposed method do not adequately address this challenge. If there is no fiber information to be found in the homogeneous patches of WM found in T1, what information is the model using? The emphasis on large receptive fields for capturing WM tissue borders indicates that one of the primary features for predicting tracking directions is a spatial point's location in the brain, and its relation to surrounding, well-known regions. However, that information is sparse relative to the level of detail found in diffusion-based tractography, and is not sufficient to predict dense fiber directions.

So, in this span of "empty" features between WM borders in a T1 image, how does the model constrain its solution space in predicting fibers? Without further evidence to the contrary, the simplest answer (as suggested in the Discussion, limitation 1) would be that the network is relying upon internal "atlases" and frequently-recurring statistical patterns found during training. This is just an *a priori* map not based upon the individual's input T1w in any way, and the immense biological variation between individuals suggests that their T1 cannot sufficiently constrain this "atlas." Also note that this is still true with the inclusion of the 122 segmentation features, as those segmentation algorithms are also given only T1 images and learned priors, and could all be learned by the student network in principle.

5. Based upon the overall motivation for this work (tractography from T1 images) and the weakness described in point 4.), it would be highly informative to compare the proposed method to a non-deep-learning streamline atlas registration method. This is also mentioned in the Introduction section, but such an approach is never tested.

6. The step size ($\gamma$) parameter is not sufficiently analyzed with respect to the proposed model performance, and there is an asymmetric use of $\gamma$ between SD_STREAM tracking and the neural network tracking. As given in the paper, the MRtrix SD_STREAM method uses the default tracking parameters, which are given in the documentation as $0.1 \times \text{voxel size} = 0.1 \times 1.25\text{mm} = 0.125\text{mm}$ (HCP diffusion data is usually 1.25mm isotropic). However, the step size used during the student network inference is 1mm, an eight-fold increase. While this may seem like a minor parameter detail, such a large difference in this parameter could drastically change the interpretation of results for at least three reasons:

- the student predicted streamlines only step 1/8th as many times as SD_STREAM, which could lead to large changes in prediction streamline paths, arcs, turns, length, etc.,
- the accumulation of error in SD_STREAM will be much larger than that of the student predictions, and
- the student network uses an RNN (GRU) which often suffer from vanishing gradients over a long enough sequence, and the length in this context is scaled inversely by the step size.

So, such a discrepancy in $\gamma$ leads to questions regarding the overall results.

**Deanonymize Review:**

no

**Detailed Comments:**

7. It is stated in the Introduction:

> "We...in a first for the field, perform tractography on T1w MRI alone..."

This seems subjective based upon the overloaded meaning of "perform tractography." As mentioned in the paper, many atlas-based methods exist that register streamlines to an individual T1w image. These methods estimate streamlines in a subject, but would this be performing tractography? Even with a more constrained definition, works such as Anctil-Robitaille, et. al. , 2022 [1] (using EuDX [2]) also trace fiber paths given only a subject's T1w image.

8. The convolution RNN/GRU does not seem particularly novel, and does not relate to the typical understanding of a CNN-RNN, which typically combines the recurrent input and convolution into a single process; there exist many examples, even within the realm of MRI processing [3, 4]. The proposed CoRNN relegates the convolution piece as a single conv layer that essentially acts as a simple spatially-informed feature expansion. Outside of this single layer, every other layer is either an MLP or GRU.

9. Figure 2 is difficult to interpret while also being the only figure to visualize a result, which makes the results overall less clear. The whole-brain tractography in a) does not provide much insight, except that the network predicted tracts seem to be "smoother." Also, it is almost impossible to find the differences between connectome maps in c), even with arrows.

10. Figure 2 the caption reads "Left" and "Right," but this is ambiguous. Are these the left and right views of the subject, or left and right partitions of the figure?

11. The scan-rescan variability of SD_STREAM seems irrelevant in relation to evaluating the proposed method. The intuition, it seems, is that scan-rescan variance in tractography is a "lower bound" for the magnitude of reasonable error for comparison of the proposed method. However, the sources of variance for scan-rescan are completely different than variance in reconstruction/prediction. The variance in scan-rescan is not even "error" in the same category as "prediction error;" comparing them side-by-side is a false equivalence.  Furthermore, the notion of scan-rescan variance as a "lower bound" is contradicted in Figure 3e, where the median graph measures of the teacher or student networks can be lower than the median scan-rescan differences.

References:

[1] B. Anctil-Robitaille, A. Théberge, P.-M. Jodoin, M. Descoteaux, C. Desrosiers, and H. Lombaert, “Manifold-aware synthesis of high-resolution diffusion from structural imaging,” Frontiers in Neuroimaging, vol. 1, 2022, Accessed: Feb. 05, 2023. [Online]. Available: https://www.frontiersin.org/articles/10.3389/fnimg.2022.930496

[2] E. Garyfallidis, “Towards an accurate brain tractography,” University of Cambridge, 2012. [Online]. Available: https://ethos.bl.uk/OrderDetails.do?uin=uk.bl.ethos.607682

[3] L. Wang, K. Li, X. Chen, and X. P. Hu, “Application of Convolutional Recurrent Neural Network for Individual Recognition Based on Resting State fMRI Data,” Frontiers in Neuroscience, vol. 13, 2019, Accessed: Feb. 06, 2023. [Online]. Available: https://www.frontiersin.org/articles/10.3389/fnins.2019.00434

[4] C. Qin, J. Schlemper, J. Caballero, A. N. Price, J. V. Hajnal, and D. Rueckert, “Convolutional Recurrent Neural Networks for Dynamic MR Image Reconstruction,” IEEE Transactions on Medical Imaging, vol. 38, no. 1, pp. 280–290, Jan. 2019, doi: 10.1109/TMI.2018.2863670.

**Paper Type:**

methodological development

**Questions To Address In The Rebuttal:**

1. The flawed basis described in point 4. above puts a large burden of proof on the authors. I would like to see a refutation of the premise laid out in 4. - that T1 images do not contain sufficient information for fiber direction estimation at the spatial scale of these data (1.25 mm isotropic), and that implies the trained student network is primarily "memorizing" fiber directions based on spatial location. This could come in many forms, including an argument from MR physics/biology, showing the network to be robust on out-of-distribution testing data (such as the lesion examples mentioned in the text), and others.

2. In relation to point 5. and question 1., I would like to understand how the registration of a streamline atlas to T1 images compares to the proposed model. This seems like a reasonable "baseline" to compare the data-driven models. How do the reconstruction metrics compare to the proposed method?

3. In relation to points 7. and 8., I would like to understand the novelty of the proposed method more fully. What is the operational definition of "perform tractography?" What is specifically novel about the CoRNN model, and in what context is it novel (all of ML, medical imaging, diffusion MRI, etc.)?

4. I would like to understand the purpose and justification of using scan-rescan variability as the lower bound for prediction error.

5. In relation to point 6., I would like to see some characterization of the step size $\gamma$ parameter being so much higher during network inference. Does this difference significantly change the results? Do the GRU layers create an inherent limitation on $\gamma$?

---

### Meta-Review · Area_Chair_1sqy · 2023-02-25

**Recommendation:** Accept (Poster)
**Confidence:** 4

**Metareview:**

This paper presents a convolutional-recurrent neural network model for learning to perform tractography from T1-weighted MRI using a teacher-student training. The strengths of this paper identified by the reviewers were (1) T1w MRI is easier and faster to acquire, so being able to avoid dMRI acquisition while still being able to perform tractography of the white matter would be advantageous, (2) the model and teacher-student paradigm are novel contributions, (3) the paper is well written and clearly organized, and (4) the results of the method were impressive and the comparison to dMRI tractography is presented well. There were two major weaknesses that stand out from the reviews: (1) One reviewer questioned the fundamental assumption behind the proposed work, namely, that there is sufficient information present in T1w MRI to accurately identify white matter directionality, given that T1w MRI does not generally provide fine-scale contrast in the white matter. This will likely be a point of controversy in the field. The authors provided additional experimental comparisons of their method versus nonlinear registration to a tractography atlas, which seem to support the additional utility of their method. (2) One reviewer noted on the rather small sample size of the test results, making statistical significance of the results difficult, and encouraged the authors to validate with larger sample sizes available from HCP. Overall, while there is real controversy of whether it is possible to infer white matter pathways from T1w MRI contrast, this paper's strengths make it a valuable contribution worthy of acceptance.